# Boosting the Performance of Artificial Intelligence-Driven Models in Predicting COVID-19 Mortality in Ethiopia

**DOI:** 10.3390/diagnostics13040658

**Published:** 2023-02-09

**Authors:** Kedir Hussein Abegaz, İlker Etikan

**Affiliations:** 1Biostatistics and Health Informatics, Public Health Department, Madda Walabu University, Robe 247, Ethiopia; 2Department of Biostatistics, Faculty of Medicine, Near East University, Near East Avenue, Nicosia 99138, Turkey

**Keywords:** artificial intelligence, COVID-19, ensemble model, adaboost, KNN, ANN-6, SVM, Ethiopia

## Abstract

Like other nations around the world, Ethiopia has suffered negative effects from COVID-19. The objective of this study was to predict COVID-19 mortality using Artificial Intelligence (AI)-driven models. Two-year daily recorded data related to COVID-19 were trained and tested to predict mortality using machine learning algorithms. Normalization of features, sensitivity analysis for feature selection, modelling of AI-driven models, and comparing the boosting model with single AI-driven models were the main activities performed in this study. Prediction of COVID-19 mortality was conducted using a combination of four dominant feature variables, and hence, the best determination of coefficient (DC) of AdaBoost, KNN, ANN-6, and SVM in the prediction process were 0.9422, 0.8618, 0.8629, and 0.7171, respectively. The Boosting model improved the performance of the individual AI-driven models KNN, SVM, and ANN-6 by 7.94, 22.51, and 8.02 percent, respectively, at the verification stage using the testing dataset. This suggests that the boosting model has the best performance for prediction of COVID-19 mortality in Ethiopia. As a result, it suggests a promising potential performance of boosting ensemble model to be applied in predicting mortality and cases from similarly recorded daily data to predict mortality due to COVID-19 in other parts of the world.

## 1. Introduction

Numerous viral disease outbreaks, including MERS, SARS, bird flu, H7N9, Ebola, H1N1, Nipah, and Zika, have occurred in the last 20 years. In December 2019, Wuhan, China, experienced the most recent coronavirus outbreak of this decade [1]. The severe acute respiratory syndrome coronavirus 2 (SARS-CoV2) is the agent of this pandemic, which has been dubbed as the coronavirus disease of 2019 (COVID-19). On March 2020, the World Health Organization (WHO) declared this illness to be a pandemic. The nature of this pandemic was different from earlier pandemic types, and had a devastating effect on the world economy and led to a nearly complete cessation of social and economic activity worldwide [1,2]. As in other parts of the world, this pandemic continues to exert impact on people in Ethiopia. It has, in fact, adversely affected the economy of the country.

COVID-19 was the cause of more than 6.7 million (6,718,464) deaths globally, as of 11 January 2023, with a case fatality rate of one percent. This rate was from early February 2020 to late December 2022. During the same period, this pandemic caused 7572 deaths in Ethiopia, with a case fatality rate (CFR) of 1.52%. This implies that the fatality rate in Ethiopia is higher than the average CFR of the world by 0.52% [3].

This pandemic proved that world was not ready to quickly control the rampant spread of such catastrophic viruses. The question is when things go back to normal. Even though no one has a final answer, we can predict what the situation would look like in the future by analyzing previously collected data. Results from the analyses will serve as knowledge for action, which could help us to manage similar pandemics in future [4,5].

Real-time disease control and easy virus tracking are both made possible by models powered by AI and machine-learning algorithms. AI has made numerous contributions to the fight against COVID-19, including prediction and tracking, contact tracing, case monitoring, early diagnosis, therapeutic drug development, vaccine development, and many others [1].

One of the most successfully recognized algorithms in the field of machine learning is the Adaptive Boosting (Adaboost) algorithm, which was developed by Freund (1997). The Adaboost algorithm, which maintains a collection of weights over training data and adjusts them after each weak learning cycle adaptively, creates a set of weak learners by assuming that a combination of weak learners can be "boosted" into an accurate strong learner [6]. In contrast to conventional back-propagation neural networks or convolutional neural networks, recent examples of research have shown that Adaboost-based machine learnings could achieve high accuracy in modeling with multi-class imbalanced data [7,8]. 

Adaboost has been used in ensemble learning because of its superior classification and prediction performance, which includes image recognition, estimation of fruit biochemical parameter, and complex change prediction modeling [9,10,11,12]. In addition to the Boosting algorithm, the prediction capability of different non-linear AI-driven superior neural network models like SOFNN-HPS and GK-ARFNN was found high in predicting wastewater treatment processes [13,14].

In Ethiopia, many studies produced information on different healthcare issues such as antenatal care (ANC) utilization status of mothers [15], the postnatal care (PNC) visit of mothers [16], access to tetanus toxoid (TT) immunization of mothers [17], predicting under nutrition status of U5 children [18], predicting the CD4 count status of patients under ART [19], predicting the level of anemia among women [20], and predicting U5 mortality [21] by using different machine-learning algorithms and AI-driven models. In addition to this, a few studies have applied them for detection and classification of COVID-19 cases from X-ray images [22,23]. In these studies, many machine learning algorithms were applied to develop their model, though many of them were classification algorithms. However, this study tried to include and test models like K-nearest neighbours (KNN), support vector machine (SVM), artificial neural network-6 (ANN-6), and the boosting algorithm (AdaBoost) to predict COVID-19 mortality (see the explanation in Section 2.4.1).

According to our search of various databases, no study has discussed the use of ensemble modeling and boosting algorithms to predict COVID-19 mortality in Ethiopia. Therefore, the objective of this study was to compare the boosting ensemble model with single and weak learner AI-driven models based on their ability to predict mortality due to COVID-19. Therefore, the main contributions of this study is that we applied a novel boosting ensemble algorithm to predict COVID-19 mortality in the Ethiopia. It compares weak learner models with the boosting ensemble model, and it provides an insight for future researchers in applying boosting algorithms to predict with higher accuracy than weak learner models allow. Hence, this study will technically contribute to the research community with the novel idea of applying boosting algorithms to daily recorded data like daily mortality of COVID-19.

## 2. Materials and Methods

### 2.1. Study Area

The study area for this specific research was Ethiopia. Ethiopia is the second-most populous country in Africa; the 2020 estimated number of population of the country was 114,964,000 [24]. Currently, the main focus under the health infrastructure development of Ethiopia is the standardization and expansion of hospitals among regional states. According to the commercial guide of the country’s report of 27 July 2022, 367 hospitals, 3777 health centers, 1,7699 health posts, 3867 private clinics, and 43 private hospitals were available, and prevention and containment of COVID-19 was first among upcoming priorities in the country [25].

### 2.2. Data Source

The COVID-19 data used in this study were those collected and archived by the Center for Systems Science and Engineering (CSSE). The dataset was available online for users by Our World in Data (OWID) and by data warehouse of John Hopkins University from the link https://github.com/owid/COVID-19-data/tree/master/public/data (accessed on 25 June 2022). OWID has published the statement under its license section saying “All visualizations, data, and code produced by ‘Our World in Data’ is completely open access under the Creative Commons License. You have the permission to use, distribute, and reproduce these in any medium, provided the source and authors are credited” [26,27]. Since the dataset used in this study does not include any personal information and was approved by the CSSE, ethical approval was not necessary.

### 2.3. Feature Selection, Data Pre-Processing, and Analysis

There were far too many variables in the COVID-19 dataset. Only seven variables, however, were taken into account for the current study because of their connection to mortality and the thoroughness of the observations. The COVID-19 daily data were non-linear by nature, so after feature selection, the first task was to normalize the chosen features after checking for normality. After conducting a sensitivity analysis with an artificial neural network (ANN), the second activity involved choosing the dominant feature variables based on their coefficient of determination (DC) values. In addition to the target variable (the daily number of COVID-19 deaths), and four feature variables (daily new cases, bed capacity, mask use, and pneumonia status) were chosen.

In order to create the weak learner AI-driven models and the boosting ensemble model (AdaBoost), the dataset was finally split into training dataset (70%) and testing dataset (30%). The tool used to normalize the data was MS-Excel. However, a powerful platform for data analysis, visualization, and modeling known as Orange (Data mining) v3.33 was utilized for the sensitivity analysis, the creation of individual AI-driven models, and the boosting model. 

### 2.4. Proposed Methods

In Figure 1, the overall proposed methodology is presented as a model development workflow in Orange (Data mining). It includes the pre-processing of data, normalizing of data, sensitivity analysis, dividing data into training and testing datasets, model development, and prediction process on both training and testing datasets. Once the data pre-processing was completed, three weak learner AI-driven models—k-nearest neighbors (KNN), the artificial neural network (ANN-6), and support vector machine (SVM)—and one boosting ensemble model (Adaptive Boosting) were developed to predict mortality due to COVID-19 in Ethiopia. Finally, the prediction performance of three AI-driven models were compared with the boosting model based on their result of the coefficient of determination (DC) and the root mean square error (RMSE).

#### 2.4.1. AI-Driven Models

The AI-driven models used in this study to predict COVID-19 mortality were the k-nearest neighbors (KNN), the artificial neural network (ANN-6), and the support vector machine (SVM). The ensemble boosting model used was the adaptive boosting (AdaBoost) AI-driven model. While developing these models, parameters selected, for each models, after trial and error is presented in Table 1 below.

Figure 2 shows the flowchart for the model development process. This flowchart demonstrates the creation of models. The data were first preprocessed in the model development process, after which individual models (KNN, SVM, and ANN-6) were developed separately and evaluated against the boosting algorithm (AdaBoost).

##### Adaptive Boosting Regression (AdaBoost Regression)

AdaBoost-based regression is a type of boosting AI-driven model that can apply a powerful machine learning algorithm for the regressing of target and feature variables [28,29,30]. The purpose of applying a boosting regression was to obtain the best prediction from the ensemble of multiple weak predictors. The schematic presentation of the AdBoost Model is presented in Figure 3.

As we can observe from the figure, the model processes the input COVID-19 dataset and lets us denote this dataset as D_x_. Initially, each dataset of D_x_ was assigned an equal weight and this weight determined the chance of being sampled. Due to this weight, the model selected the training dataset (D_x1_) from dataset D_x_ with replacement sampling, and hence, to train the regressor *f_1_(x)*, the training dataset was used. 

As we can see from the schematic presentation, the prediction weight assessment was applied to assess the trained regressor 1 [*f_1_(x)*] and calculate the weight ‘w_1_’ for the regressor. This assessment is to adjust the weight for the main dataset D_x_. In the weighting process, the larger the prediction error, the larger the weight for that specific trained dataset. Finally, the model parameter used in this study to build the AdaBoost model after a lot of trial and error was (Base estimator: *tree*, Number of estimators: *4*, Algorithm: *Samme.r*, and Loss (regression): *Square*). (See Table 1)

Min H and Luo, X. (2016) have summarized the overall procedure of AdaBoost in eight steps, which is presented as follows [31]:

Step 1: The dataset D_x_ with training samples can be represented as ${\left\{(x_{j},y_{j})\right\}}_{j=1}^{M}$.

Step 2: To assign equal distribution of weight it can be presented as $\left\{p_{ij}=\frac{1}{L}|i=1,2,\ldots K;j=1,2,\ldots M\right\}$ for each training samples starts from i = 1 and starts the loop.

Step 3: In the ith iteration, the sample training data (M) from ${\left\{(x_{j},y_{j})\right\}}_{j=1}^{M}$ will be replaced with $p_{ij}$, and the sampled data are used to train a regressor $g_{i}(x;\beta_{i})$.

Step 4: Calculate the prediction loss $L_{j}=L[y_{j,}g_{i}(x_{j};\beta_{i})]$ for each member of D_x_, where $L_{j}\in [0,1]$. In addition, calculate the weighted average of the loss $\bar{L}$.
$$L_{j}=\frac{L[y_{j,}g_{i}(x_{j};\beta_{i})]}{D},D=\sup\{L_{j}\},j=1,2,\ldots$$
$$\bar{L}=\sum_{j=1}^{M}p_{ij}L_{j}$$

Step 5: The weight of the regressor $g_{i}(x;\beta_{i})$will be calculated, and it can be presented by the following formula:$$w_{i}=\frac{\bar{L}}{1-\bar{L}}$$

Step 6: If ‘i’, in step 5, equals the maximum number of iteration K, it will stop the loop and move to step 8.

Step 7: Updating the distribution weight of the dataset D_x_ by making i = i + 1 in equation L_j_ at Step 4 and moving to the following loop: $$p_{ij}=\frac{p_{ij}w_{i}^{1-\bar{L}}}{Z_{i}}$$
where Z_i_ is a selected normalized factor and, hence, P_ij_ will be a random distribution.

Step 8: The obtained K regressors will be incorporated into a single regressor respective to their weight ${\left\{w_{i}\right\}}_{i=1}^{K}$ and it will have the folowing formula:$$g(x;\beta_{\mathrm{weight}})=\sum_{i=1}^{K}w_{i}g_{i}(x,\beta_{i})$$

##### K-Nearest Neighbors Regression (KNN Regression)

KNN regression is one of the best-known and simplest non-parametric regression types and it does not explicitly assume the parametric form of the target variable [32]. Given a prediction point of X_0_ and the value for K, the KNN regression will first identify the K training observations, which are closest to X_0_, represented by N_0_. The KNN then estimates the target variable Y using the average of all the training responses in N_0_. The small number of K provides the most flexible fit that has a low bias but high variance, and hence, the optimum value for K will depend on the bias-variance tradeoff.

We can present the prediction formula of KNN as follows: $Y=\frac{1}{K}\sum_{x_{i}\in N_{0}}y_{i}$

In this study, the model parameters for the KNN were decided after much trial and error of the model development process, and hence the parameters that make KNN to predict better than other parameters were (Number of neighbors: *2*, Metric: *Manhattan*, and Weight: *Uniform*) (See Table 1).

##### The Artificial Neural Network (ANN-6)

As it can establish a connection between feature variables and the target variable by training neural networks without detailed knowledge of the dataset, the ‘ANN-6’ is a class of AI-driven model and is regarded as the most important model [33]. In a variety of scientific fields, including biomedicine, technology, agriculture, and business, ANN is more effective and useful [34]. This is because of its self-learning simulation function, which shows how ANNs can predict and model complex processes like the daily number of COVID-19 mortality. In order to predict COVID-19 mortality, the ANN-6 with forward propagation algorithm was chosen. The model parameters of the ANN-6 were Hidden layers: *200*, Activation: *tanh*, Solver: *L-BFGS-B*, Alpha: *1*, Max iterations: *500*, and replicable training: *True*) (See Table 1).

The ANN-6 with the Broyden-Fletcher-Goldfarb-Shanno (BFGS) optimization algorithm and with three layers (input layer, hidden layer, and output layer) was selected after repeated trial and error assuming different parameters with optimum prediction capability. In addition to the trial and error, the BFGS has a proven performance even for non-smooth optimization [35] like the daily mortality of COVID-19. 

##### The Support Vector Machine (SVM)

The SVM is an AI-driven model and supervised machine learning algorithm type designed for classification and regression [2,36,37]. The regression of SVM was applied to predict using the regression known as support vector regression. Before applying the SVM, it was important to select the kernel function. Hence, this study used the radial basis function (RBF) kernel type after training the model with 70% of the data to predict mortality due to COVID-19 in Ethiopia, by combining all feature variables. The performance ability of RBF is better than the rest of the kernel types (sigmoidal and polynomial) and RBF has fewer turning parameters than others [2,38], hence, we prefer to model the SVM by using the RBF. The final parameters of the SVM model after a trial and error was (SVM type: *SVM, C = 1.0, ε = 0.10000000000000003*, Kernel: *RBF, exp(-auto|x-y|^2^)*, Numerical tolerance: *0.001, and* Iteration limit: *300*) (See Table 1).

#### 2.4.2. Data Normalization and Model Performance Evaluation 

Before modelling the AI-driven models and the boosting model, the standardization of the target variable and feature variables was conducted to normalize the data into the standardized value ranging from 0 and 1. This standardization assures reducing dimensions among variables and having equal attention in the modelling process [2,39]. If the variable to be normalized is ‘X’, the normalization formula will be as follows: $$X_{n}=\frac{x_{i}-x_{\min}}{x_{\max}-x_{\min}},i-1,2\ldots .n$$
where $X_{n}$, $x_{i}$, $x_{\min}$, and $x_{\max}$represent the normalized, the actual, the minimum, and the maximum value of the variable X.

The coefficients of determination (DC) and root mean square error (RMSE) were computed in the performance evaluation of models. Based on the determined values of RMSE and DC, the top performing model was chosen. Therefore, a model with a DC value close to 1 and the lowest RMSE was deemed to be the best-performing model. 

## 3. Results and Discussion

Three AI-driven models (KNN, ANN-6, and SVM) and one boosting model (AdaBoost) were modelled for this study. All these models were trained on 70% of the COVID-19 dataset and tested on 30% of this dataset. This section included successive reports on feature statistics, sensitivity analysis, the creation of AI-driven models, and a comparison of those models to the boosting model. 

### 3.1. Feature Statistics

The minimum, mean, maximum, and standard deviation (SD) values of the target and feature variables are presented in Table 2 below for both training and testing datasets. The average number (mean ± SD) of daily mortality due to COVID-19, between 01 April 2020 and 01 April 2022, was (9.13 ± 8.21) for the training dataset and (13.27 ± 12.78) for the testing dataset. The average number of daily cases was (604.08 ± 539.79) for the training dataset and (756.02 ± 1063.06) for the testing dataset. In addition to daily deaths and daily cases, the average bed capacity per/1000, the daily mask use (measured from 1), and the pneumonia status were (0.17 ± 0.02), (0.42 ± 0.16), and (0.96 ± 0.09), respectively, for the testing dataset.

The radar chart described the daily number of new deaths in Figure 4. In this chart, four largest numbers of daily mortality due to COVID-19 were 49, 48, 47, and 47 deaths on 28 September 2021, 1 October 2021, 9 September 2021, and 20 April 2021, respectively. In addition to the largest number of daily deaths, 38 and more daily deaths were registered in the country from 13 September 2021 to 14 October 2021. Therefore, we can summarize that the peak time of COVID-19 mortality in Ethiopia was from 1 April 2020 to 1 April 2022.

### 3.2. The Sensitivity Analysis

To obtain the optimum level of prediction of AI-driven models, the most important step is to carefully select the most relevant feature variables and to adjust model parameters for every model. In the sensitivity analysis, seven variables were included. These were ‘mask_use’, ‘all_bed_capacity’, ‘new_cases’, ‘pneumonia_st’, ‘icu_bed_capacity’, ‘hosp_admission’, and ‘daily_infection’. In previous classical models, the linear sensitivity analysis techniques were applied to select the dominant feature variables. However, the daily recorded data related to COVID-19 have a non-parametric nature. Hence, the neural network sensitivity analysis (the FFNN) was conducted to choose the dominant feature variables and is presented in Table 3.

As we can observe from Table 3, four variables scored a DC value greater than 0.5, and accordingly, ‘mask_use’, ‘all_bed_capacity’, ‘new_cases’, and ‘pneumonia_st’ were ranked from first to fourth, respectively, and were used to build all models in this study. However, those feature variables with DC value less than 0.5 were excluded from the model building.

### 3.3. Prediction of COVID-19 Using Single AI-Driven Models

In the modelling process, the data were trained and tested by using three AI-driven models (KNN, SVM, and NN) and one boosting model (AdaBoost). Hence, the prediction performance of each model is presented in Table 4. 

The model that we applied in this study to boost the prediction performance of COVID-19 in Ethiopia was the AdaBoost model. In this model, a variant called “AdaBoost.samme.r” was applied. This variant works with classifiers that can show output prediction probabilities. The values of DC and RMSE obtained, from testing dataset, were 0.9422 and 2.0549, respectively. This implies that the AdaBoost model was the best performer in predicting COVID-19 mortality in Ethiopia.

The second AI-driven model used to predict COVID-19 mortality was the KNN. In this model, both assumptions of weight (uniform and distance) were tried in the modeling process. However, the KNN with ‘distance’ weight was going to be over-fitted and the KNN with ‘uniform’ weight was best-fitted. Therefore, the values of DC and RMSE were 0.8618 and 3.1858, respectively. Accordingly, the KNN was the third-best performer model to predict COVID-19 in Ethiopia, next to the AdBoost and the ANN-6.

The third AI-driven model used in this study to predict COVID-19 was SVM. To build the SVM model using selected dominant feature, the kernel of the radial basis function (RBF) was applied. This function was selected due to its better performance than that of the other types of functions under the SVM in predicting COVID-19 in eastern Africa [2]. As presented in Table 4, the performance of SVM in predicting COVID-19 was reported in the form of DC and RMSE, whereby the value of the DC was 0.7171 and that of the RMSE was 4.5461 in the testing dataset. This result implies that the prediction performance of SVM was less than that of the other prediction models.

The fourth AI-driven model used was the ANN-6. The Broyden-Fletcher-Goldfarb-Shanno (BFGS) optimization algorithm was selected due to its proven performance, even for non-smooth optimization [26]. This implies that the ANN-6 was a good predictor for non-linear data such as the COVID-19 daily mortality. The value of DC was 0.8620 and that of the RMSE was 3.1749 in the testing dataset, implying that the ANN-6 was the second-best performer algorithm to predict COVID-19 deaths in Ethiopia, next to the boosting algorithm and the first AI-driven algorithm among three single and weak learner models.

### 3.4. The Correlation Analysis

The relationship between the actual and the predicted values of daily mortality due to COVID-19 using four AI-driven models (AdaBoost, KNN, SVM, and ANN-6) was calculated and presented in Figure 5. In this visual presentation, the rank of AI-driven models in predicting COVID-19 mortality was presented in bivariate correlation values. Hence, the correlation values were 0.9706, 0.9289, 0.9283, and 0.8468 for AdaBoost, ANN-6, KNN, and SVM, respectively. This implies that AdaBoost, ANN-6, KNN, and SVM were the first, second, third, and fourth models, respectively, to indicate fewer spread points in the correlation with mortality due to COVID-19 in Ethiopia, thereby producing a better estimated value of the mortality.

The scatter plot in Figure 5 and the results of the AI-driven models’ analysis in Table 4 helped us understand that the boosting algorithm outperformed the others in terms of predicting COVID-19 mortality in Ethiopia. According to this finding, the AdaBoost algorithm was the most effective AI-driven model for predicting COVID-19 data that are gathered on a daily basis. 

A bivariate correlation analysis using the spearman correlation coefficient was conducted and the result is presented in Figure 6. In this analysis, the observed value of daily mortality was correlated with each observed feature variable and each predicted value from the AI-driven models (AdaBoost, KNN, ANN-6, and SVM). The predicted values with AdaBoost, ANN-6, KNN, and SVM algorithms were the first, second, third, and fourth highly correlated ones, with values of 0.971, 0.931, 0.929, and 0.867, respectively. In addition to this, mask use, all bed capacity, and daily new cases were the first three highly correlated feature variables with values of 0.873, 0.796, and 0.765, respectively. However, the pneumonia case was the lowest correlated feature variable. Hence, we understood from this result that the spearman correlation value was improved among AI-driven models.

### 3.5. Comparison of AdaBoost with Single AI-Driven Model 

Table 5 compares the boosting model with AI-driven models in terms of prediction performance across training and testing datasets. In a training dataset, the AdaBoost model improved the prediction accuracy of KNN, SVM, and ANN-6 models by 8.48%, 22.31%, and 8.96%, respectively. Additionally, it improved the accuracy of prediction for KNN, SVM, and ANN-6 models in a testing dataset by 7.94%, 22.51%, and 8.02%, respectively. The results indicated that ensemble boosting models could be used to predict COVID-19 mortality in Ethiopia more effectively than the tested single AI-driven models. 

### 3.6. The Taylor’s Diagram

We can visualize the performance of various AI-driven models in a single diagram known as Taylor’s Diagram to quickly comprehend it. This diagram, which is shown in Figure 7, is a two-dimensional diagram that coordinates the standard deviation (SD) and correlation coefficient (r) of each AI-driven model’s predicted value (using AdaBoost, KNN, ANN-6, and SVM) and the observed values of COVID-19 mortality. The significance of using this diagram is that it quantifies the degree of similarity between the predicted values and the observed values of mortality while simultaneously displaying the predicting performance of various models in a single visual display. Figure 7 makes it clear that "AdBoost" was the AI-driven model that performed the best in predicting COVID-19 mortality in Ethiopia, with (r = 0.9706 and SD of 0.0907), and that the SVM was the model that performed the worst, with (r = 0.8468 and SD = 0.0934).

## 4. Conclusions and Recommendations

In this study, the prediction performance of the boosting model was compared to that of the single AI-driven models in an investigation into how well they predicted COVID-19 mortality. The performance of single AI-driven models in predicting COVID-19 mortality was investigated and the boosting model was compared with the single AI-driven models in terms of performance of prediction. Before commencing prediction, data were normalized, and a sensitivity analysis was conducted to select dominant feature variables. Finally, three single AI-driven models and one boosting model was developed to predict mortality, and the prediction performance of these three models was compared with that of the boosting model. At the verification stage using the testing dataset, AdaBoost boosted the prediction of performance of three models KNN, SVM, and ANN-6 models in a testing dataset by 7.94, 22.51, and 8.02 percent, respectively.

Overall, findings of this study demonstrated the boosting ensemble model’s capacity to predict COVID-19 mortality in Ethiopia was much higher than the single and weak learner AI-driven models. This shows that boosting ensemble models have a promising ability to predict COVID-19-related mortality in other parts of the world and to apply this model to predict mortality and other cases from similarly recorded daily data. Furthermore, this study used only two years of daily recorded COVID-19 mortality and other feature variables to develop the single models and the boosting model. Therefore, it is important to test these AI-driven boosting models for further data with a large number of observations in future studies.

## Figures and Tables

**Figure 1 diagnostics-13-00658-f001:**
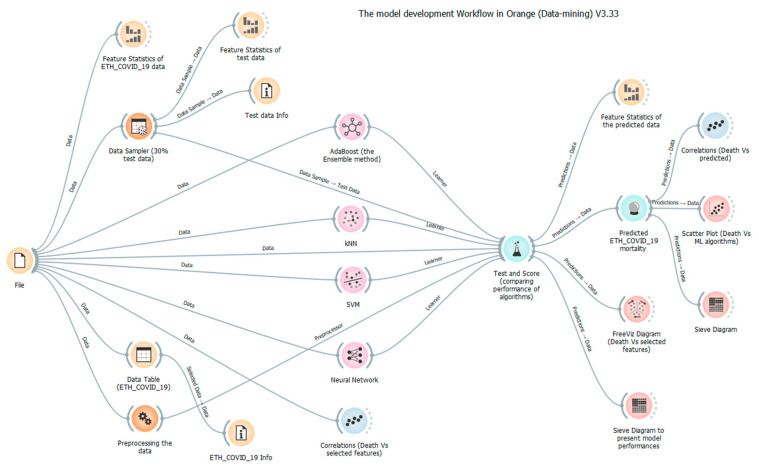
Orange (Data mining) workflow of the proposed methodology.

**Figure 2 diagnostics-13-00658-f002:**
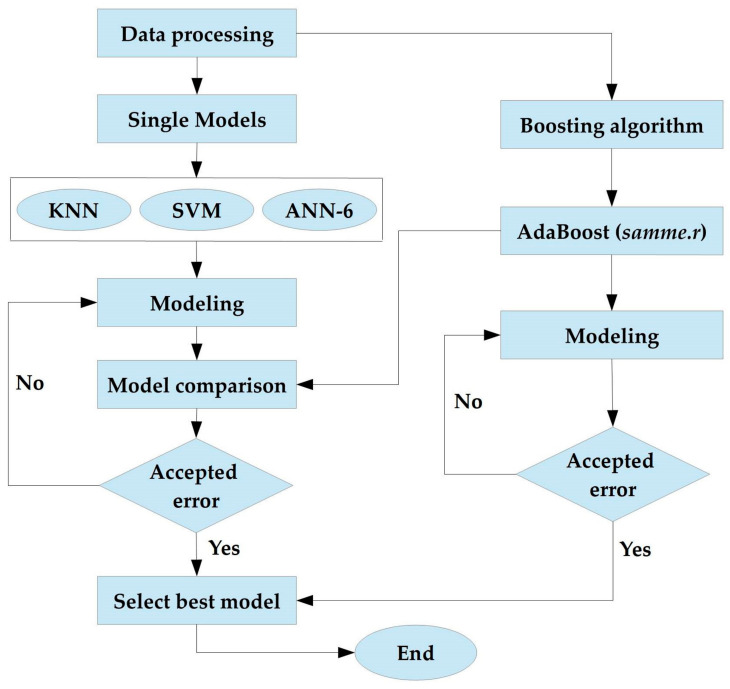
Flowchart of model developments.

**Figure 3 diagnostics-13-00658-f003:**
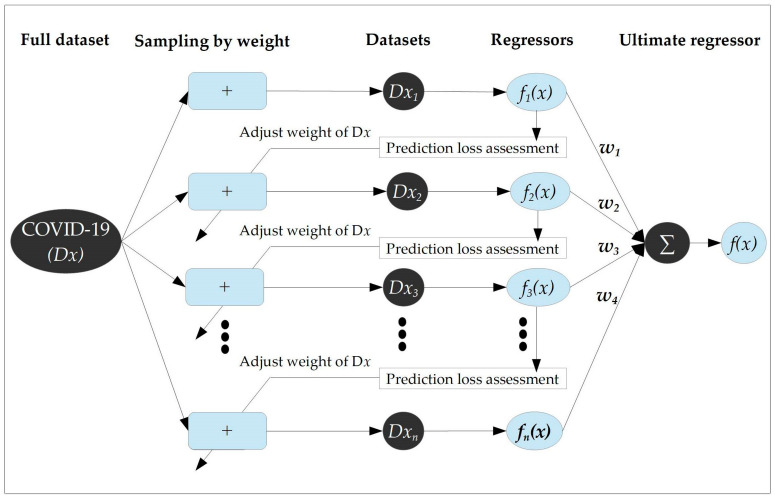
Schematic diagram of AdaBoost regression.

**Figure 4 diagnostics-13-00658-f004:**
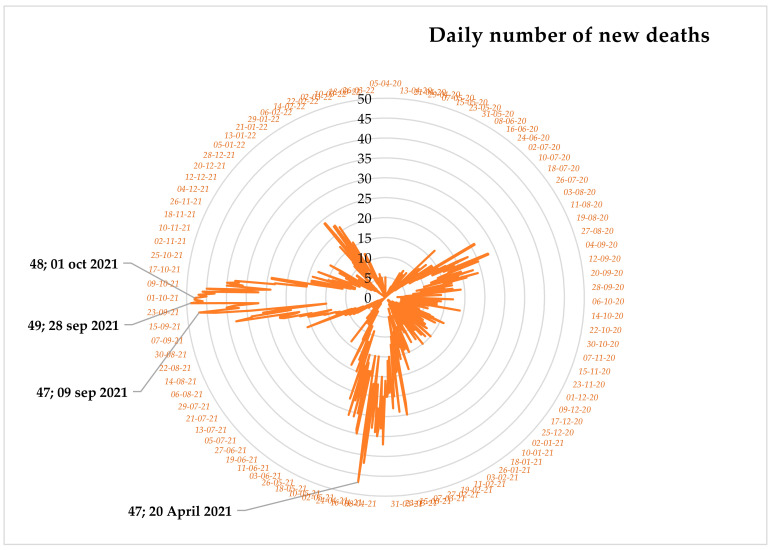
Radar chart for daily COVID-19 mortality.

**Figure 5 diagnostics-13-00658-f005:**
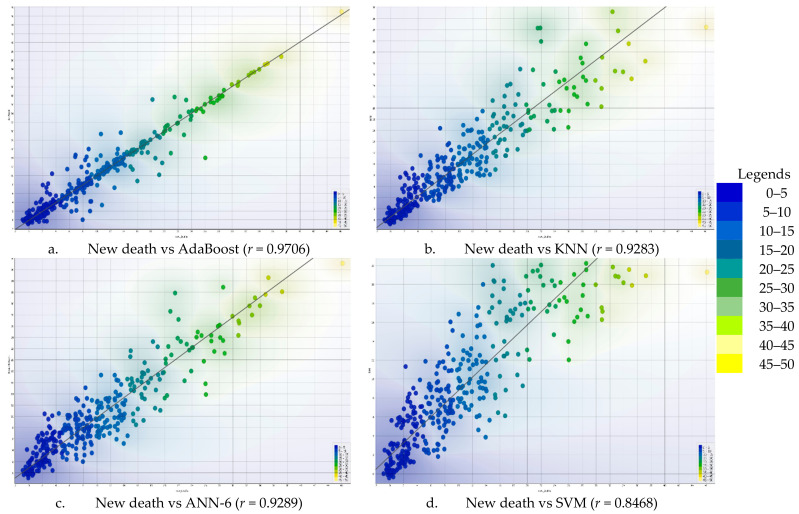
Correlation between the predicted and the actual values of COVID-19 mortality using AI-driven models.

**Figure 6 diagnostics-13-00658-f006:**
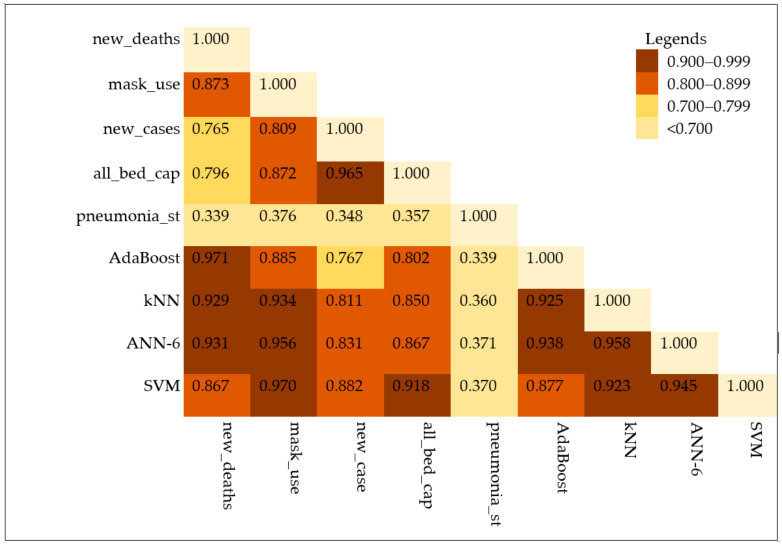
Correlation statistics among the input variables and the predicted mortality.

**Figure 7 diagnostics-13-00658-f007:**
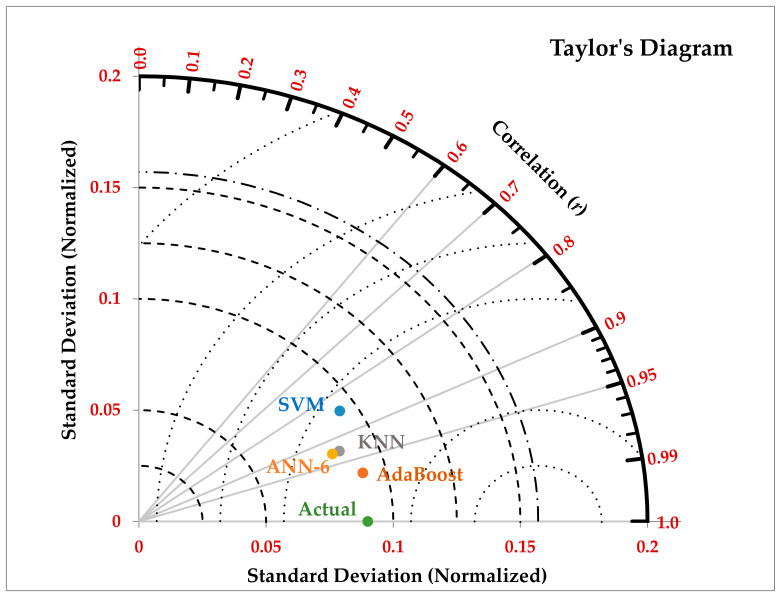
Taylor’s diagram showing the prediction performance of models.

**Table 1 diagnostics-13-00658-t001:** Model parameters used to build AI-driven model criteria.

AI-Driven Models	Model Parameters
AdaBoost	Base estimator: *tree*, Number of estimators: *4*, Algorithm: *Samme.r*, and Loss (regression): *Square*
KNN	Number of neighbours: *2*, Metric: *Manhattan,* and Weight: *Uniform*
SVM	SVM type: *SVM, C = 1.0, ε = 0.10000000000000003*, Kernel: *RBF, exp(-auto|x-y|^2^)*, Numerical tolerance: *0.001, and* Iteration limit: *300*
ANN-6	Hidden layers: *200*, Activation: *tanh*, Solver: *L-BFGS-B*, Alpha: *1*, Max iterations: *500*, and Replicable training: *True*

**Table 2 diagnostics-13-00658-t002:** Descriptive statistics of target and feature variables of COVID-19 dataset.

Variables	Training Dataset (n = 584, 70% of the Data)			Testing Dataset (n = 146, 30% of the Data)		
	Mean ± SD	Min	Max	Mean ± SD	Min	Max
New deaths	9.1298 ± 8.2090	0	47	13.2667 ± 12.7766	0	49
New cases	604.0812 ± 539.7915	0	2372	756.019 ± 1063.059	7	5185
Bed capacity	0.1647 ± 0.0235	0.1245	0.1856	0.1729 ± 0.0213	0.1345	0.1741
Mask use	0.4279 ± 0.1633	0.0000	0.6689	0.4163 ± 0.1641	0.0000	0.8679
Pneumonia_st	0.9615 ± 0.0961	0.8213	1.0929	0.9629 ± 0.0963	0.8132	1.1294

**Table 3 diagnostics-13-00658-t003:** Sensitivity analysis applied to select dominant feature variables.

Features Included	Longer Description of Feature Variables	DC	Rank
‘mask_use’	Percent of population reporting always wearing a mask	0.867	1st
‘all_bed_capacity’	Total number of beds that exists at the location	0.815	2nd
‘new_cases’	Daily number of new cases	0.796	3rd
‘pneumonia_st’	Ratio of pneumonia deaths to the average annual deaths	0.768	4th
‘icu_bed_capacity’	Total number of ICU beds that exists at the location	0.421	5th
‘hosp_admission’	Daily COVID-19 hospital admission	0.401	6th
‘daily_infection’	The number of daily infections	0.253	7th

**Table 4 diagnostics-13-00658-t004:** AI-driven models to predict COVID-19 in Ethiopia using the combination of the selected four dominant features.

Model	Feature Combinations	Model Parameters	Training Dataset		Testing Dataset	
			RMSE	DC	RMSE	DC
AdaBoost	mask, all_bed, cases, pneumonia	Samme.r	1.9358	0.9449	2.0549	0.9422
KNN	mask, all_bed, cases, pneumonia	Uniform	3.0834	0.8601	3.1858	0.8618
SVM	mask, all_bed, cases, pneumonia	RBF	4.3482	0.7218	4.5461	0.7171
ANN-6	mask, all_bed, cases, pneumonia	L-BFGS-B	1.9358	0.8553	3.1749	0.8629

**Table 5 diagnostics-13-00658-t005:** Comparison of boosting model with weak learner AI-driven models.

Boosted Model vs. Single Model	Difference in Percentage	
	Training Dataset	Testing Dataset
AdaBoost vs. KNN	8.48%	7.94%
AdaBoost vs. SVM	22.31%	22.51%
AdaBoost vs. ANN-6	8.96%	8.02%
KNN vs. SVM	13.83%	14.57%
KNN vs. ANN-6	0.48%	0.08%
ANN-6 vs. SVM	13.35%	14.49%

## Data Availability

The data used for the analysis in this study are available from K.H.A. upon request.

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
