# Peer review of "Boosting the Performance of Artificial Intelligence-Driven Models in Predicting COVID-19 Mortality in Ethiopia"

_diagnostics, 2023, doi:10.3390/diagnostics13040658_

Round 1

Reviewer 1 Report

1. Rewrite the Abstract more precisely by explaining the purpose of study; methodology used; major findings; summary of your interpretations and implications.

2. In INTRODUCTION section, discuss the problem background, specific terms related to the study, and purpose of the research work..

3. The exact technical contribution and novelty of the work need to be stated explicitly.

4. Interpretation and analysis of experimental observations are extremely important - authors must pay proper attention to this in the manuscripts.

5. Discuss the experimental results by paying attention to novelty and effectiveness of the proposed approach.

6. Write a concise CONCLUSION, describing overall work carried out, and future scope of the study.

7. Manuscripts need to be free from grammatical and other language errors.

8. Update some of the references with latest papers (2020-2023)

Reviewer 2 Report

1. It is recommended to further strengthen the introduction of the significance of the study in the introduction section.

2. It is recommended that the introduction to the existing methodology be further strengthened in the introduction section.

3. It is suggested to compare with some superior neural network-based algorithms, such as SOFNN-HPS, GK-ARFNN, in the introduction section of the intelligent algorithm.

4. It is suggested to present the pseudo-code or flowchart of the algorithm.

5. It is suggested to further point out the future improvement trend of the algorithm or the direction of the research in the conclusion section.
